# Methanolic Extracts of *D. viscosa* Specifically Affect the Cytoskeleton and Exert an Antiproliferative Effect on Human Colorectal Cancer Cell Lines, According to Their Proliferation Rate

**DOI:** 10.3390/ijms241914920

**Published:** 2023-10-05

**Authors:** Chiara Anglana, Makarena Rojas, Chiara Roberta Girelli, Fabrizio Barozzi, Josefa Quiroz-Troncoso, Nicolás Alegría-Aravena, Anna Montefusco, Miriana Durante, Francesco Paolo Fanizzi, Carmen Ramírez-Castillejo, Gian-Pietro Di Sansebastiano

**Affiliations:** 1Department of Biological and Environmental Sciences and Technologies (Di.S.Te.B.A.), University of Salento, Via Monteroni, 73100 Lecce, Italy; chiara.anglana@unisalento.it (C.A.); makarena.rojas@unisalento.it (M.R.); chiara.girelli@unisalento.it (C.R.G.); fabrizio.barozzi@unisalento.it (F.B.); anna.montefusco@unisalento.it (A.M.); fp.fanizzi@unisalento.it (F.P.F.); 2Oncology Group IDISSC and Biomedical Technology Centre (CTB), Biotecnology-B.V. Departament ETSIAAB, Universidad Politécnica de Madrid, 28223 Madrid, Spain; josefa.quiiroz@gmail.com (J.Q.-T.); nicolas.alegria.aravena@gmail.com (N.A.-A.); carmen.ramirez@ctb.upm.es (C.R.-C.); 3Deer Production and Biology Group, Regional Development Institute, University of Castilla-La Mancha, 02006 Albacete, Spain; 4Institute of Sciences of Food Production (ISPA-CNR), 73100 Lecce, Italy; miriana.durante@ispa.cnr.it

**Keywords:** *D. viscosa*, colorectal cancer, extract, antiproliferative effect, cytoskeleton, tubulin, tonoplast

## Abstract

Numerous studies have reported the pharmacological effects exhibited by *Dittrichia viscosa*, (*D. viscosa*) including antioxidant, cytotoxic, antiproliferative, and anticancer properties. In our research, our primary objective was to validate a prescreening methodology aimed at identifying the fraction that demonstrates the most potent antiproliferative and anticancer effects. Specifically, we investigated the impact of various extract fractions on the cytoskeleton using a screening method involving transgenic plants. Tumors are inherently heterogeneous, and the components of the cytoskeleton, particularly tubulin, are considered a strategic target for antitumor agents. To take heterogeneity into account, we used different lines of colorectal cancer, specifically one of the most common cancers regardless of gender. In patients with metastasis, the effectiveness of chemotherapy has been limited by severe side effects and by the development of resistance. Additional therapies and antiproliferative molecules are therefore needed. In our study, we used colon-like cell lines characterized by the expression of gastrointestinal differentiation markers (such as the HT-29 cell line) and undifferentiated cell lines showing the positive regulation of epithelial–mesenchymal transition and TGFβ signatures (such as the DLD-1, SW480, and SW620 cell lines). We showed that all three of the *D. viscosa* extract fractions have an antiproliferative effect but the pre-screening on transgenic plants anticipated that the methanolic fraction may be the most promising, targeting the cytoskeleton specifically and possibly resulting in fewer side effects. Here, we show that the preliminary use of screening in transgenic plants expressing subcellular markers can significantly reduce costs and focus the advanced characterization only on the most promising therapeutic molecules.

## 1. Introduction

Plants are widely recognized for their role as a natural source of remedies for various diseases, including cancer [1,2]. However, the complex composition of plant extracts can sometimes obscure the identification of the most effective compounds, as some may have lower efficacy or unintended side effects [3,4]. Therefore, it is essential to investigate each plant extract to uncover compounds with targeted antiproliferative effects while minimizing off-target cytotoxicity, a challenge shared with conventional chemotherapeutic agents. Off-target cytotoxic effects may be so severe that they cause serious intoxication [5].

Some chemotherapeutic molecules block signal transduction, inhibit growth-promoting signals from classic endocrine hormones (hormonal therapies), or inhibit growth-signals like those associated with receptor tyrosine kinases (targeted therapy). Nonetheless, the most effective chemotherapeutics inhibit cell division [6] or induce DNA damage [7]. Another promising strategy relies on targeting the cytoskeleton.

*Dittrichia viscosa* is a perennial plant distributed in the hillsides, waste grounds, and marshy areas of different regions of the Mediterranean Basin. It has many ethnomedical uses, including anti-inflammatory, anthelmintic, lung disorders, antipyretic, antiseptic, and antiphlogistic activities [8,9], in addition to treating gastroduodenal disorders [10]. Many studies have reported that *D. viscosa* exhibits pharmacological effects, including antioxidant effects, cytotoxic effects, antiproliferative, and anticancer effects [11,12,13].

Colorectal cancer ranks among the most prevalent cancers, affecting both genders. It ranks second in females, after breast cancer, and third in males, following lung and prostate cancers. In patients with metastasis, the primary objective of chemotherapy is to extend survival and maintain a good quality of life. However, its effectiveness has been limited by severe side effects and by the development of resistance [14]. For this reason, additional therapies are needed in order to fight the problem of selectivity and drug resistance. Medicinal plants are highly interesting for this purpose, but the discovery of candidate molecules rich in phytocomplexes needs to be accelerated and more efficient. Working in this direction, efficiency can be increased by lowering the cost of preliminary screening and reducing later characterization efforts.

Here, we show the specific effect of a specific *D. viscosa* extract fraction on the cytoskeleton using a transgenic plant screening method [15,16]. This cytotoxic effect demonstrated promising antiproliferative activity on colon-like cell lines characterized by the expression of gastrointestinal differentiation markers (e.g., HT-29 cell line) and undifferentiated cell lines, showing the positive regulation of epithelial–mesenchymal transition and TGFβ signatures (e.g., DLD-1, SW480, and SW620 cell lines). These diverse cell lines were selected to represent the heterogeneity observed in colon tumors. Considering that colorectal cancer cell lines serve as accurate molecular models of primary carcinomas for preclinical studies, these different tumor lines can provide valuable insights into varying molecular responses to cancer treatments. All three *D. viscosa* extract fractions exhibited antiproliferative effects, with the prescreening of transgenic plants indicating that the methanolic fraction might be the most promising due to its potential to preferentially target microtubules, the critical components of the mitotic bundle in proliferating cell lines.

## 2. Results

### 2.1. Chemical Characterization

Three fractions were prepared [17] from *D. viscosa* phytocomplex, namely hexanic extract (HE), methanolic extract (ME), and aqueous extract (AE), with each characterized for the relative content of compounds of general interest.

Soluble phenolic content varied from 11.37 to 168.16 mg of Gallic Acid Equivalent/g of dried extract (mg GAE/g dw). The highest concentration (168.16 ± 6.83 mg GAE/g dw) was found in AE; also, flavonoids were more abundant in AE with a value of 207.54 ± 2.91 mg of Catechin Equivalent/g of dried extract(mg CE/g dw). Other biomolecules, such as total vitamin C have the highest values in AE with 15.11 ± 0.19 mg/g, while the higher value of condensed tannins was in HE with a value of 8.04 ± 0.17 mg CE/g. The methanol extract was midway between the other two extracts. The content of other biomolecules is shown in Table 1.

The three extracts were also tested for their antioxidant capacity with the highest values in ME and AE 1503.22 ± 18.89 and 2116.76 ± 30.75 µmol Trolox equivalents/g, respectively (Table 2).

### 2.2. ^1^H-NMR Characterization of Dittrichia Viscosa Extracts Spectra

The HE, ME and AE of *D. viscosa* were characterized by 600 MHz ^1^H-NMR spectroscopy. The visual inspection of the ^1^H-NMR spectra revealed the presence of different metabolites (Figure 1, Figure 2 and Figure 3). The assessment of the metabolites observed in the ^1^H-NMR spectra was also performed on the basis of the ^1^H−^1^H COSY and ^1^H−^13^C (HSQC and HMBC) 2D spectra and confirmed with literature data [18,19,20,21].

In the HE, characteristic signals ascribable to sesquiterpenes tomentosin and α-costic acid were identified. These compounds are known to be the main component of *D. viscosa* [22], and their properties are well known (Figure 1).

In particular, the spectral information for the sesquiterpene lactone tomentosin:methyl protons (position 14) appear as a doublet at 1.12 ppm, and the diagnostic singlet at 2.14 was assigned to the methyl group in position 15. Multiplets at 3.30 and 4.64 ppm were assigned to the H7 and H8 proton, respectively. The two doublets identified at 6.26 and 5.52 ppm were ascribed to the H13 protons of lactone ring.

Moreover, as confirmed by carbon HSQC (125 ppm) and HMBC correlations (40, 146 and 171 ppm), the presence of two intense correlated singlets at 6.32 and 5.69 ppm was assigned to the two protons (H13_α_ and H13_β_) of the olefinic methylene group in eudesmane-type sesquiterpens, α-costic acid. The assignment was confirmed by the presence of methyl singlets at 0.82 ppm (position 14) and 1.60 ppm (position 15), with respective couplings observed in the COSY, HSQC, and HMBC bidimensional spectra. The broad singlet at 5.32 ppm (121 ppm ^13^C) was assigned to proton in position 3. Among sesquiterpene lactones, as already reported in the literature, signals of the inuviscolide trycyclic skeleton were observed [23]. In particular, the two broad singlets at 5.10 and 4.97 ppm were ascribed to the exomethylene protons (position 14), as confirmed by HSQC and HMBC experiments. Indeed, the methyl group at position 15 appeared as a singlet observed at 1.19 ppm (21 ppm, ^13^C). Finally, characteristic signals of fatty acids were identified. Terminal methyl groups (*CH_3_*) of all fatty acid chains (SFA, MUFA, and DUFA) and ω3 PUFA were observed at 0.88 ppm and at 0.98 ppm, respectively, as confirmed by HMBC assignment (14 ppm). Intense signals in the range 1.26–1.29, 1.65–1.77 and 2.24 ppm were identified as characteristic for methylene (n-*CH_2_*), β-methylene (–*CH_2_*CH_2_COO–), and α-methylene (–*CH_2_*COO–) protons of all fatty acids chains. The unsaturated fatty acids (UFA) were identified by the diagnostic resonances of allylic (–*CH_2_*CH=CH–) and olefinic (–*CH*=*CH*–) protons at 2–2.10 and 5.36 ppm, respectively. The presence of polyunsaturated fatty acids (PUFA), such as linoleic (C18:2 ω6) and linolenic (C18:3 ω3), was assessed by the presence of bis-allylic protons (–CH=CH*CH_2_*CH=CH–) at 2.85 ppm, as already described in the literature [24]. Finally, among several signals between 3.5 and 4.6 ppm, the resonances at 4.17 and 4.35 ppm were assigned to the glycerol 1–3 protons of triacylglycerols. Moreover, the signals at 8.20 and 8.16 indicate the presence of flavonoid derivatives, as reported in the literature [25]. Finally, carotenoids and chlorophylls, were identified in the range between 6.4 and 6.6 ppm, and 9.26 and 9.64 ppm, respectively [26].

For ME (Figure 2), signals of metabolites already identified in the hexanic extracts, such as fatty acid chains and sesquiterpenes, were observed. Diagnostic resonances for hydroxycinnamic acids were observed in the spectral range of 6.7–7.10 ppm. As already observed for the hexanic extract, signals from carotenoids and chlorophylls were identified in the spectral region between 6.4 and6.6 ppm, and 9.26 and 9.64 ppm, respectively Interestingly, the presence of intense signals in the range of 11 to 13.5 ppm indicate the protons of a hydroxyl group in position 5 (5-OH), characteristic to flavonoids, as previously described [27].

For AE (Figure 3), in the aliphatic region, signals at 1.90–2.00 correlated with carbon, and those at 39.5 ppm were assigned to the quinic acid moiety of hydroxycinnamic acids. The signals of the anomeric protons of common sugars α/β glucose and sucrose were observed at 5.24/4.65 and 5.42, respectively. In the aromatic region characteristic signals for the caffeoyl moieties of hydroxycinnamic acids were observed. In particular, as reported in the literature [28], chlorogenic, neo chlorogenic, and dicaffeoylquinic acids were identified in *D. viscosa* extracts. For chlorogenic acid: 5.34 (H5, m); 6.39 (H8′, d, J = 15.9); 6.93 (H5′, d, J = 8.1); 7.12 (H6′, dd J = 8.4, 2.1); 7.19 (H2′, d J = 2.1); 7.65 (H7′, d J = 15.9). Dicaffeoylquinic acid: 6.48 (H8″, d J = 15.6); 7.65 (H7′, d J = 16.2). And for neochlorogenic acid: 6.37 (H8′, d J = 15.9); 7.66 (H7′, d J = 15.9). More deshielded singlets were assigned to nucleotides derivatives (8.15 and 8.20 ppm) and formate (845 ppm).

### 2.3. HPLC and GC-MS Analysis

HPLC analysis was performed for the identification and quantification of polyphenols, tococromanols, and carotenoids. Among the polyphenols that were most abundant in the AE and ME, chlorogenic acid (22.90 mg/g in AE), di-O-caffeoylquinic acid and its isomer (99.5 mg/g in ME and 106.67 mg/g in AE), and rosmarinic acid (7.77 mg/g in AE) were identified. Tococromanols and carotenoids were present at moderate concentration only in the HE (1764.44 µg/g and 1278.98 µg/g, respectively). The only carotenoid that was present in the ME was lutein (227.53 µg/g), (Table 3).

A different percentage of fatty acid, characterized with GC-MS, was in the HE and ME extracts, with both extracts showing a higher concentration of polyunsaturated fatty acids (PUFA) than saturated fatty acids. This was more accentuated in ME, with 67.03% of PUFA. Among saturated fatty acids, (SFA) myristic acid was more abundant in the HE (13.14%), as well as palmitic acid (22.67% in the HE), which also had a high percentage in the ME (18.59%). Among the polyunsaturated fatty acids (PUFA) the most abundant was linolenic acid (28.49% in the HE and 49.99% in the ME) (Table 4).

### 2.4. Screening of Extracts on Transgenic Cells in the Plant Hypocotyl

The effects of extracts on the biology of the cell were investigated on transgenic plants expressing GFP-tagged proteins evidencing important cellular characteristics GFP-TUA6 to monitor cytoskeleton [15,16,17,18,19,20,21,22,23,24,25,26,27,28,29] and GFP::SYP51 to monitor the endomembranes of the secretory system [30,31]. These markers label important subcellular structures that are differently organized in plant cells but based on molecular processes largely connected with animal and human cells [15].

GFP-TUA6, a GFP-tagged α-tubulin TUA6 that integrates into endogenous microtubules, labels the distribution of this cytoskeleton component (Figure 4A,B). Normally microtubules are distributed in a network with a variable angle, but in this case, the observed effect consists of their depolymerization. The negative control DMSO, a plant extract solvent, did not influence the normal pattern after 18 h application (Figure 4C). AE also had no evident effect on MTs pattern, even at the highest dose of 3 g/L (Figure 4D).

The HE applied at 350 mg/L had no evident effect after 1 h (Figure 4E) but was lethal after 18 h (Figure 4F), causing the collapse of cells. Also, when applied at 750 mg/L, the HE had initially no effect on the cytoskeleton (Figure 4G) but became lethal over time (Figure 4H).

The ME behaves differently. When applied at 450 mg/L, it had no evident effect after 1 h (Figure 4I), but it disorganized MTs over time and completely destructured the cytoskeleton after 18 h (Figure 4J). Increasing the ME concentration to 1 g/L showed an immediate effect on the cytoskeleton within 1 h (Figure 4K), which gradually worsened over time (Figure 4L).

GFP-tagged AtSYP51, GFP::SYP51, is a TGN and tonoplast marker. It was sorted as the related QcSNARE without the fluorescent tag and sorted as a transmembrane protein to the tonoplast. The normal tonoplast pattern shows the large perimeter of the central vacuole (Figure 5A). This is not altered by DMSO (Figure 5B) nor by the AE applied for a short time (Figure 5C). The AE is somehow able to induce a strong stress within 18 h, as evidenced by the distribution of GFP in small dots, which is possibly related to marker retention in the TGN or in multivesicular compartments (Figure 5D).

The HE affected the tonoplast pattern immediately both at low (Figure 5E) and high doses, causing the solubilization of GFP in the vacuole in the surviving cells after 18 h (Figure 5F). This effect is probably due to a strong induction of autophagy and stress related multivesicular body formation since these are the processes able to internalize and solubilize a membrane-anchored protein in the vacuoles, such as GFP::SYP51 [30].

Again ME showed a different behavior. No effect was observed after a short exposure of cells to low or high doses (Figure 5G) even if the high dose was able to alter the cytoskeleton in the short period (Figure 4K). A fluorescent pattern alteration was only observed after longer exposure time (Figure 5H), possibly as the result of traffic impairment to cytoskeleton disorganization. The relation between microtubules and autophagy in plants is complex [32], but their disruption induces a cell death with a deep perturbation of endomembranes.

### 2.5. Effect on Human Colorectal Cancer Cell Lines

In our study we used colon-like cell lines characterized by the expression of gastrointestinal differentiation markers (such as the HT-29 cell line) and undifferentiated cell lines showing the positive regulation of epithelial–mesenchymal transition and TGFβ signatures (such as the DLD-1, SW480, and SW620 cell lines). *D. viscosa* extracts (hexanic, methanolic, and aqueous extract) have had a qualitatively and quantitatively different effect on the different cell lines. The extracts in hexane solvent had greater efficacy in all the cell lines (comparative scale of cell viability and IC50 in Figure 6A–F). The strangest effect and the most significant differences were observed in the metastatic SW620 line. At 50 µg/mL, the hexanic extract’s effect over the HT-29 cell culture was statistically higher than the effect in DLD-1 cells, which were described to be more resistant to these treatments with MSI (genomic instability phenotypes microsatellite instability), MSH5 down-regulated, and the up-regulation of the c-kit gene expression. This statistical difference was also observed in the SW480 line, which is microsatellite stable, or in the HT-29 cell line (Figure 6A,D).

The methanolic solvent extracts appeared to show fewer differences in the response of the tumor lines to treatment. Again, a greater effect can be observed in the SW620, HT-29 and DLD-1 lines, as regards decreased proliferation (lower expression of Ki67 Table 5), but the effect on SW620 was seen particularly early (Figure 6B,E).

Finally, the aqueous extract showed the greatest difference in the effect caused between lines. DLD-1 and HT-29 were more resistant to the effect of aqueous extracts. Statistically significant differences were recorded at concentrations of 1 mg/mL of extract and higher. The SW480 cell line, which is a primary tumor as DLD-1 and HT-29 lines, was also more resistant to treatment than the SW620 line from a metastasis (Figure 6C,F).

## 3. Discussion

The chemical analysis of *D. viscosa* extract aligns with the existing literature, confirming the presence of high levels of sesquiterpenes and flavonoids. These compounds are believed to contribute to the extract’s anticancer properties. Numerous studies have highlighted the biological activities of sesquiterpene lactones, particularly their cytotoxic, anti-tumoral, and anti-inflammatory properties [33,34,35]. Tomentosin and inuviscolide were shown to be present in the hexane extract or in the ethanol and ethyl acetate extracts, showing cytotoxic effects against cancer cell lines and triggering apoptosis overcoming drug resistance in tumor cells [14,36,37]. These compounds could inhibit tumor growth through the selective alkylation of key enzymes, which are regulators of cell division, causing cells apoptosis [38].

Flavonoids and chlorogenic acids, which are typically present in methanol, aqueous, and ethanol extracts, have demonstrated antioxidant, antiproliferative, anticancer, anti-inflammatory, and antimicrobial properties. The methanol extract (ME) consistently displayed greater effectiveness compared to a water extract [11,39,40,41,42,43,44].

Molecules normally found in ME, in addition to chlorogenic acids, include nepetin, hispidulin and methylated quercetins [45], kampferol, quercetin and coumarin [41], catechin, luteolin and ferulic acid [42]. Our methanolic extract also showed higher levels of flavonoids and higher antioxidant abilities, which are likely attributable to these molecules.

In those cases with no available literature to support the discovery of anticancer metabolites, it is easy to detect antiproliferative effects of phytocomplexes, but the discovery of the most effective compounds remains a time-consuming task. Here, we investigate the specific effect of each *D. viscosa* extract fraction on the membranous compartmentalization of the cells and cytoskeleton organization by observing the distribution of GFP-tagged proteins at a subcellular level, as previously proposed by other studies [15,16]. We selected the marker GFP-TUA6 [15,29] because it labels the microtubules integrating into endogenous α-tubulin TUA6, and GFP::SYP51 [30,31] because it labels not only the tonoplast of the central vacuoles, but also the intermediate compartments of the vacuolar sorting pathway. GFP-TUA6 thus shows the distribution of microtubules and the immediate cytoskeleton alterations, while a GFP::SYP51 alteration only occurs after important changes in membrane traffic and cellular compartmentalization, such as in apoptosis.

These two markers help to evaluate the nature of the cytotoxic effects and of the antiproliferative effect of a given molecule. An early effect on the cytoskeleton with no evident effect on endomembranes should indicate an antiproliferative effect due to mitotic microtubule bundles disruption without strong cytotoxicity on other cellular mechanisms. On the contrary, GFP::SYP51 redistribution would imply compartmental rearrangement and a deep alteration of traffic, with strong effects on the cells’ housekeeping activities.

Tumors are heterogeneous, and targeting the cytoskeleton offers a promising approach with fewer off-target effects. Tubulin is a crucial molecular target for antitumor drugs due to its role in mitotic spindle formation [46]. Inhibiting microtubule formation through tubulin inhibitors often induces apoptosis [47,48] and is less likely to lead to multi-drug resistance (MDR) compared to other metabolites [49,50].

Our pre-screening analyzed the three extract fractions at low and high concentrations. The low dose was chosen because it was able to induce diversified effects on tumor cell lines, while the high dose was chosen at the limit of practical use, as it was able to eventually overcome problems related to the plant cell wall barrier. We found that the ME had a quite evident effect on microtubules polymerization without significantly affecting cell compartmentalization in the short term. The effect on endomembranes derives from cytoskeleton disruption and required time, leading to cell death in plants, and we hypothesize the same effect in tumor cells. In other words, when cytoskeleton remodeling is observed first, the endomembranes alteration is a consequence leading to PCD. This means that the greatest effect would be on proliferating cells affecting the mitotic bundle. If endomembranes are altered before or at the same time as the cytoskeleton, the molecular target of the extract is unpredictable and may easily cause off-target effects in human cells.

High doses of the ME have a clear effect on the cytoskeleton within 1 h (Figure 4K) without affecting endomembranes (Figure 5G). On the contrary, the HE has no effect on the cytoskeleton even at high doses (Figure 4G), but it strongly affects endomembranes (Figure 5E,F).

Tumor heterogenicity is clear, and in all patients, it is influenced by different factors, such as the microenvironment or epigenetics. Also, colon tumors are molecularly different, and the colorectal cancer lines we used are representative of this heterogenicity. Based on the view of CRC cell lines as accurate molecular models of primary carcinomas for preclinical studies, the different tumor lines can provide information on the different molecular behavior of response to tumor treatments. In our study we have used colon-like cell lines characterized by expression of gastrointestinal differentiation markers (such as the HT-29 cell line) and undifferentiated cell lines showing the positive regulation of the epithelial–mesenchymal transition and TGFβ signatures (such as the DLD-1, SW480, and SW620 cell lines). At the DNA level, this includes the genomic instability phenotypes, microsatellite instability (MSI) (DLD-1 cell line), and the epigenomic CpG island methylator phenotype (CIMP) (DLD-1 and HT-29 cell lines). About 15% of primary CRCs have MSI, while the rest are microsatellite stable (MSS) (SW480 and SW620 cell lines), most of which have the chromosomal instability phenotype (CIN).

*D. viscosa* extracts (hexanic, methanolic, and aqueous extract) have had a qualitatively and quantitatively different effect on the different cell lines, possibly due to their differential mechanisms of resistance to anoxic processes, oxidative stress conditions, and apoptotic evasion. A pleiotropic effect of the extracts cannot be ruled out, nor, above all, a non-specific cell-death-inducing effect affecting healthy cells as well as tumor cells, just like the cytostatics normally used as chemotherapeutics. As with current chemotherapy, this effect is more damaging to dividing cells, hence the applicability of neoplastic therapies against rapidly dividing abundant cells characteristic of the tumor mass, and the certain protection recorded for slower dividing cells. This also applies to our extracts, but pre-screening on transgenic plants helps to increase the chance to focus research on compounds targeting the cytoskeleton first.

Greater efficacy was observed in all the cell lines treated with the HE. As for the molecular mechanisms that may be regulating this different cellular response to treatments, we are faced with the interesting signaling pathway of the epithelial–mesenchymal transition (EMT) that is active in cells of high tumorigenicity, as it seems to affect our extracts. This signaling pathway is a great challenge and a great therapeutic target. In the SW620 line, which is most affected in our assays, there is a decrease in MicroRNA 145 (miR-145) which targets multiple stem cell transcription factors and with an expression that is inversely correlated to the EMT. miR-145 has been raised as a therapeutic target to reverse radiation resistance (RT) mediated by the Snail family transcriptional repressor 1 (SNAI1) [51]. Therefore, in future works, we will address the study of the regulation of the expression of both the SNAI1 factor and its regulator miR-145 to see if this could be the pathway that is being affected in our cells by the SNAI1 factor. To reiterate the importance of these data, in two different already-found types of solvents of the same extract, it is interesting to note that these molecules are more effective in inhibiting the proliferation and survival of highly tumorigenic and proliferative cells. At 50 µg/mL, the HE effects over HT-29 cell culture were statistically higher than the effects in DLD-1 cells, more resistant to these treatments with MSI (genomic instability phenotypes microsatellite instability), and described and down-regulated MSH5, with up-regulation of the c-kit gene expression. The same differences were observed in the microsatellite stable SW480 line as in the HT-29 cell line. Other differences between these lines are two mutations present only in DLD-1 cells in the protein PI3KCA that could be involved in their resistance to hexanic extract components.

The other strong effects caused by the HE may be due to the disruption of a number of cellular processes that may represent a higher risk of off-target effects on patients cells. In fact, the HE has been shown to have strong cytotoxic power on tumor cell lines, but in the screening system with transgenic plants, it was lethal for the cell, causing the collapse of cells’ endomembranes at low concentrations.

The ME showed less evident differences in the response of the tumor lines to treatment. But again, greater effects can be observed in the SW620, HT-29, and DLD-1 lines, with respect to the less proliferative (lower expression of Ki67, Table 5). Moreover, the effect on SW620 was extremely rapid and at lower doses. Since SW620 cells are the most actively proliferating, we can hypothesize a correlation between the effect of the ME on plant cell microtubules organization and the stronger effect on SW620.

Finally, the AE showed the greatest differences in the effect caused between lines, but doses were high. Cells with extensive methylation in the CpG islands (CIMP), such as DLD-1 and HT-29, were more resistant to the effect of the aqueous extracts. Statistically significant differences were recorded at concentrations of 1 mg/mL of extract and higher. This epigenetic modification may be conditioning a greater resistance and anti-apoptotic capacity of the transcriptome of these cells. The SW480 cell line is a primary tumor, as are the DLD-1 and HT-29 lines, which may condition a response to treatments that is more different from the SW620 line or from a metastasis, as shown in Figure 6. This could have implications in the applications derived from a future treatment with this type of extract, which would lead to greater efficacy in advanced metastatic tumors, specifically those against which we currently lack effective treatments. It would also be very important to test this type of treatment in the future in cell lines of other types of tumors that are more resistant to current treatments, such as glioblastoma multiforme, one of the most aggressive tumor types, since any advances in the treatment of this type of tumor can be considered of great scientific import and translational advance in clinical practice.

In conclusion, all three extract fractions have antiproliferative effects, but the pre-screening on transgenic plants anticipated that the ME may be the most promising for reduced off-target effects. This multidisciplinary study showed that the simple visual screening of extracts’ effects on plant tissues expressing specific fluorescent markers can be used to search for interesting natural phytocomplexes, better characterize the subcellular effects of new drugs, focus research efforts, and reduce the costs of drug discovery. This is our third use-case validation of the use of transgenic plants as support in the characterization of subcellular effects of complexes or single molecules [15,16], and we believe the method [52] is ready for new challenges in real pre-screening.

The next step of this specific work will be the study of synergies combining chemotherapy with plant extracts in order to study the possibility of improving the response of patients with treatments by decreasing the dose of chemotherapy, thereby potentially eliminating side effects and maintaining the expected anti-neoplastic effect thanks to the extracts of *D. viscosa.* The limitations of the study focus on the pleiotropic effect that plant extracts can have, which must be studied on the basis of effective doses and combinations that allow for effective synergies. On another front, we can consider a wider range of eluents that improve the extractions of *D. viscosa* or provide new properties not yet highlighted, such as NADES (natural eutectic solvents that can optimize the extraction of polyphenols and antioxidant power in plant preparations).

## 4. Materials and Methods

### 4.1. Plant Extracts

*D. viscosa* plants were collected from population DI3 present at Campus Ecotekne (40°20′00.18″ N 18°07′02.33″ E). The aerial parts of the non-flowering plants were frozen, freeze-dried, ground in a mill, and extracted with different solvents of increasing polarity, namely n-hexane, methanol, and water, following the procedure described by [17]. Three fractions were obtained identified as follows: hexanic extract (HE), methanolic extract (ME), and aqueous extract (AE). 50 mg of each extract were solubilized in 5 mL of the corresponding solvent used for the extraction. From each solution, 1 mL was taken and dried on a rotavapor. The dry residue of each sample was resuspended in 1 mL of the specific solvent for the chemical analysis to be carried out. All samples were centrifuged at 3000 rpm for 2 min, and the supernatant was recovered to prepare three diluted replicates for each sample to perform chemical tests.

### 4.2. Determination of Soluble Phenols

The total phenol content of the three extracts was measured using the Folin–Ciocalteu method described by [53] with some modifications. The extracts were resuspended in 1 mL of 80% ethanol. 450 µL of distilled H_2_O and 50 µL of Folin–Ciocalteu reagent were added to 50 µL of each sample or 50 µL of 80% ethanol for the blank. After five minutes, 500 µL of a 7% sodium carbonate solution and 200 µL of distilled H_2_O were added, bringing each sample to a final volume of 1250 µL. After 90 min in the dark at room temperature, the samples were transferred into cuvettes to be read in the UV-VIS spectrophotometer (UV-2600 Shimadzu, Columbia, MD, USA) at a wavelength of 750 nm. To determine the amount of phenols present in the samples, a calibration curve was constructed using known concentrations (1–2–4–8–10–12 µg/100 µL) of gallic acid (Sigma-Aldrich, St. Louis, MO, USA) in 80% ethanol. The soluble phenol content was expressed as mg of gallic acid equivalents/g of dry plant weight (mg GAE/g dw).

### 4.3. Total Flavonoid Content (TFC)

The quantification of TFC was carried out according to the method described by [54]. The dry residue of extract was resuspended in 1 mL of 100% methanol. 50 µL of each sample and 50 µL of 100% methanol for the blank were diluted with 450 µL of distilled H_2_O. Then, 30 µL of 5% NaNO_2_ was added, followed by 60 µL of 10% AlCl_3_ after 5 min. After incubation for another 6 min, 200 µL of NaOH 1M and 210 µL of distilled H_2_O were added. Subsequently, the samples were vortexed and transferred into cuvettes to be read using the spectrophotometer at a wavelength of 510 nm. To determine the amount of flavonoids present in the samples, a calibration curve was constructed using known concentrations (400–200–100–50–25–12.5–6.25–3.125 µg/mL) of catechin (Sigma-Aldrich, St. Louis, MO, USA) in 100% methanol. The results were expressed in mg catechin equivalents/g of dry plant weight (mg CE/g dw).

### 4.4. Total Tannins Content (TTC)

The TTC was determined using the method of vanillin in an acid medium described by [55].

600 µL of a 4% vanillin (Thermo Fisher Scientific, Waltham, MA, USA) solution in methanol was added to 100 µL of sample (resuspended in 1 mL of 100% methanol) and 100 µL of 100% methanol for the blank. Then, 300 µL of concentrated HCl were added and left to stand for 15 min at room temperature in the dark. Finally, samples were transferred to cuvettes and read using a spectrophotometer at a wavelength of 500 nm. To determine the amount of condensed tannins, present in the samples, a calibration curve was constructed using known concentrations (250–125–62.5–31.25–15.62–7.81–3.90 µg/mL) of catechin in 100% methanol. The results were expressed in mg catechin equivalents/g of dry plant matter (mg CE/g dw).

### 4.5. Total Vitamin C Content

Ascorbic acid (AsA) and dehydroascorbic acid (DAsA) were determined by the method of [56], with slight modifications. The dry residue of each sample was resuspended in 1 mL of 6% TCA. The assay involved the use of a series of reagents added to 50 µL of sample or blank (6% TCA). For AsA content: 150 µL of phosphate buffer 0,2 M pH 7.4, 50 µL of distilled H_2_O, 250 µL of 10% TCA, 200 µL of 42% orthophosphoric acid, 200 µL of 4% 2,2’-dipyridyl (Sigma-Aldrich, St. Louis, MO, USA) in 70% ethanol and 100 µL of 3% FeCl_3_. For AsA + DAsA: 50 µL of DTT 10 mM buffer, 100 µL of phosphate buffer 0.2 M pH 7.4, 50 µL of 0.5% NEM (Sigma-Aldrich, St. Louis, MO, USA), 250 µL of 10% TCA, 200 µL of 42% orthophosphoric acid, 200 µL of 4% 2,2’-dipyridyl in 70% ethanol and 100 µL of 3% FeCl_3_. At the end of the reactions, the samples were transferred into cuvettes and were read using the spectrophotometer at a wavelength of 525 nm. The standard curve for the determination of vitamin C was constructed using solutions at known concentrations (600–500–400–300–200–100 μM) of ascorbic acid in 6% TCA.

### 4.6. Total Chlorophylls and Carotenoids Content

The total content of chlorophylls and carotenoids was carried out following the method of [57]. The dry residue of each sample was resuspended in 5 mL of 100% acetone.

The quantification of the different pigments, expressed in µg/mL, was carried out, reading at wavelengths of 661.6 nm, 644.8 nm and 470 nm using the spectrophotometer and using the following mathematical formulas:

Chlorophyll a = 12.25 A_661.6_ − 2.79 A_644.8_

Chlorophyll b = 21.5 A_644.8_ − 5.1 A_661.6_

Total carotenoids = (1000 A_470_ − 1.82 Ca − 85.02 Cb)/198

Where A = absorbance, Ca = chlorophyll a concentration, and Cb = chlorophyll b concentration.

### 4.7. Evaluation of Antioxidant Activity

The TEAC method (Trolox Equivalent Antioxidant Capacity) described by [58] was used to determine the antioxidant activity. For this assay, the monocationic radical ABTS^∙+^ was produced starting from a solution of ABTS (diammonium salt of 2,2′-azinobis-3-ethylbenzothiazolin-6-sulphonic acid) (Sigma-Aldrich, St. Louis, MO, USA) 7 mM to which was added potassium persulphate K_2_S_2_O_8_ 2.45 mM. The solution was left under stirring in the dark at room temperature for 16 h. To test the activity of the water-soluble antioxidants in the methanolic and aqueous extracts, to 10 µL of sample we added 1 mL of ABTS^∙+^ diluted in PBS 5 mM pH 7.4 (1:90, *v*/*v*). The blank consisted of 1 mL of PBS 5 mM pH 7.4, while the Trolox solution consisted of 1 mL of ABTS^∙+^ diluted in PBS 5 mM pH 7.4 + 10 µL of methanol or 10 µL of water. To test the activity of fat-soluble antioxidants, 10 µL of the Hexanic extract was added to 1 mL of ABTS^∙+^ diluted in ethanol (1:90, *v*/*v*). The blank consisted of 1 mL of ethanol, while the Trolox solution 0 µM consisted of 1 mL of ABTS^∙+^ diluted in ethanol + 10 µL of hexane.

The samples were left at room temperature for 15 min, then were read spectrophotometrically at a wavelength of 734 nm. Two calibration curves were constructed using known concentrations (15–10–7.5–5–2.5–0 µM) of Trolox (6-hydroxy-2,5,7,8-tetramethylcroman-2-carboxylic acid) (Sigma-Aldrich, St. Louis, MO, USA) in PBS (Phosphate-Buffered Saline) 5 mM pH 7.4 (to determine the activity of water-soluble antioxidants) and in ethanol (to determine the activity of fat-soluble antioxidants). The antioxidants activity was expressed as µmol Trolox equivalents/g of dry plant matter.

### 4.8. ^1^H-NMR Spectroscopy

Spectra acquisition was performed at 300 K using a Bruker Avance III 600 Ascend NMR spectrometer (Bruker, Ettlingen, Germany) operating at 600.13 MHz for ^1^H observation and equipped with a TCI cryoprobe incorporating a z-axis gradient coil and automatic tuning matching (ATM).

For *D. viscosa* hexanic and methanolic extracts: a total of 600 μL of deuterated chloroform (CDCl3) containing 0.03 *v*/*v*% TMS (sodium salt of trimethylsilyl propionic acid) as a chemical shift reference was added to (5 mg) of the dry extract and transferred to a 5 mm NMR tube. A one-dimensional experiment (zg Bruker pulse program) was run with 64 scans, 64 K time domain, spectral width 20.0276 ppm (12,019.230 Hz), 2 s delay, p1 8 μs, and 2.73 s acquisition time.

For *D. viscosa* aqueous extract: a total of 600 μL of deuterium oxide (D_2_O) containing 0.05% *w*/*v* TSP-d4 (sodium salt of trimethylsilyl propionic acid) as a chemical shift reference was added to (5 mg) of the dry extract and transferred to a 5 mm NMR tube.

A one-dimensional experiment with pre-saturation and composite pulse for selection (zgcppr Bruker standard pulse sequence) was acquired, with 128 scans, 16 dummy scans, 5 s relaxation delay, a size of fid of 64 K data points, a spectral width of 12,019.230 Hz (20.0276 ppm), and an acquisition time of 2.73 s.

The resulting FIDs were multiplied by an exponential weighting function corresponding to a line broadening of 0.3 Hz before Fourier transformation, automated phasing, and baseline correction. Metabolite identifications were based on ^1^H and ^13^C assignment by 1D and 2D homo- and heteronuclear experiments (2D ^1^H Jres, ^1^H COSY, ^1^H-^13^C HSQC, and HMBC) and by comparison with the literature data [20,23,24,25,26,27,59]. NMR data processing was performed by using TopSpin 3.6.1 (Bruker, Biospin, Milano, Italy).

### 4.9. HPLC Analysis of Polyphenols

Extracts were analyzed for polyphenol content, as reported in [60], using an Agilent 1100 Series HPLC system (Agilent Technologies, Santa Clara, CA, USA) equipped with a Phenomenex-luna 5 μm C18 (2) 100 Å column (250 × 4.6 mm), (Phenomenex, Torrance, CA, USA), and the temperature of the column was set at 30 °C. The flow rate of the mobile phase was 1.0 mL/min, and the injection volume was 20 μL. A gradient elution program was utilized with a mobile phase consisting of acetonitrile (solution A) and water solution H_3_PO_4_ (10 mL/L) (solution B) as follows: isocratic elution, 100% B, 0–30 min; linear gradient from 100% B to 85% B, 30–55 min; linear gradient from 85% B to 50% B, 55–80 min; linear gradient from 50% B to 30% B, 80–82 min; post time, 10 min before the next injection. The wavelengths used for quantification of phenolic compounds were 280, 295, and 320 nm. Peaks were identified by comparing their retention times and UV–Vis spectra to that of standards.

### 4.10. HPLC Analysis of Isoprenoids (Tocochromanols, Carotenoids, and Chlorophylls)

The dried extracts were suspended in 100 μL ethyl acetate and assayed, qualitatively and quantitatively as in [61] using an Agilent 1100 Series HPLC system equipped with a reverse-phase C30 column (5 µm, 250 # 4.6 mm) (YMC Inc., Wilmington, NC, USA). To record the HPLC runs, the Agilent ChemStation Rev A 10.02 software was used. The mobile phases were methanol (A), 0.2% ammonium acetate aqueous solution/methanol (20/80, *v*/*v*) (B), and tert-methyl butyl ether (C). The gradient elution was as follows: 0 min, 95% A and 5% B; 0–12 min, 80% A, 5% B, and 15% C; 12–42 min, 30% A, 5% B, and 65% C; 42–60 min, 30% A, 5% B, and 65% C; and 60–62 min, 95% A and 5% B. The column was re-equilibrated for 10 min between runs. The flow rate was 1.0 mL/min, and the column temperature was maintained at 25 °C. The injection volume was 10 µL. Absorbance was registered at 290 nm for tocopherols, 475 nm for carotenoids, and 675 nm for chlorophylls. Peaks were identified by comparing their retention times and UV–Vis spectra to those of authentic isoprenoid standards.

### 4.11. GC-MS Analysis of Fatty Acids

Five of each extracts were evaporated to dryness under a stream of nitrogen. The derivatization of fatty acids was carried out according to [62]. Three mL of 0.5 M NaOH (dissolved in methanol) was added to dried extracts. The mixture was incubated at 100 °C for 5 min in a water bath to dissolve lipids. After cooling at room temperature, 2.0 mL of boron trifluoride in methanol (12% *w*/*v*) was added, and the sample incubated at 100 °C for 30 min in a water bath and then rapidly cooled in an ice bath before the addition of 1 mL of hexane for extraction. The sample was vigorously stirred for 30 s before the addition of 1 mL of a 0.6% *w*/*v* sodium chloride solution. After centrifugation (6000× *g*, 2 min at 4 °C), the organic upper phase was recovered and analyzed by GC/MS analysis, as described in Durante et al. (2016), using an Agilent 5977E GC/MS system equipped with a DB-WAX column (60 m, 0.25 mm i.d., 0.25 mm film thickness) (Agilent Technologies, Santa Clara, CA, USA). The GC parameters were as follows: the temperature of the column was 50 °C after injection for 1 min, then programmed at 25 °C/min to 200 °C, at 3 °C/min to 230 °C, and maintained at constant temperature of 230 °C for 23 min. Split injection was conducted with a split ratio of 5:1, the flow-rate was 1.0 mL/min, the carrier gas used was 99.999% pure helium, the injector temperature was 250 °C, and the column head pressure was 40 psi for 0.4 min, with a constant pressure of 20 psi. The MS detection conditions were as follows: transfer line temperature 250 °C, mode Scan, source and quadrupole temperature 230 °C and 150 °C, respectively. The scanning method of acquisition, ranging from 46 to 500 for mass/charge (*m*/*z*), was optimized. Spectrum data were collected at 0.5 s intervals. Solvent cut time was set at 2 min, and the retention time was set at 40 min, which is sufficient for separating all the fatty acids. Compounds were identified by using the online NIST-library spectra and published MS data. Moreover, fatty acids standard was used to confirm MS data.

### 4.12. Cell Lines and Cell Culture

Colorectal cancer cells (DLD-1, HT-29, SW480, and SW620) were procured from the American Type Culture Collection (ATCC, Manassas, VA, USA). Table 5 shows the main differential characteristics of the colon cancer lines used in this study.

Cells lines were cultured in Dulbecco’s modified Eagle’s medium (DMEM-High Glucose, Dominique Dutscher, Bernolsheim, France), 10% fetal bovine serum (FBS, PAN Biotech, Aidenbach, Germany), 2 mM L-glutamine (FBS, PAN Biotech, Germany), and 1% penicillin/streptomycin (Corning, New York, NY, USA) at 37 °C in a humidified incubator (Series II water Jacker, Thermo Scientific, Waltham, MA, USA) with 5% CO_2_.

### 4.13. Dose–Response Tests

Cells were seeded on 96-well plates (p-96) (Deltalab S.L, Barcelona, Spain) at a concentration of 200,000 cells/mL and were left in the incubator for 24 h at 37 °C with 5% CO_2_. After, the compounds to be evaluated were added at 400, 500, 600, 1000, 1500, 2000, 2500, and 3000 µg/mL, along with the negative control, to the aqueous extract; 30, 60, 90, 180, 360 µg/mL, along with the negative control, to the hexanic extract; and 50, 100, 150, 200, 300, 375, 450, 500, 600 µg/mL, along with the negative control, to the methanolic extract over the cells, and were left to incubate for 72 h at 37 °C with 5% CO_2_. Each experiment has four replicates per condition in each plate, with four independent experiments in total.

### 4.14. Cell Viability Assay

A thiazolyl blue tetrazolium bromide assay (MTT, BioChem, PanreacApplichem, Barcelona, Spain) was used to detect the viability after 72 h. The reactive was added in each well to 0.5 mg/mL MTT solution.

The plate was incubated for 4 h at 37 °C with 5% CO_2_. After, it was emptied, and DMSO was added to solubilize tiazol precipitations (Labbox Labware, S.L., Barcelona, Spain). The detection of absorbance was read using the spectrophotometer BIOBASE-EL10A (Biobase, Jinan, China) at 546 nm.

### 4.15. Transgenic Plants and Confocal Microscopy

Chimerical GFP-tagged markers were stably expressed in *A. thaliana*, as previously described. GFP-tagged α-tubulin TUA6 (GFP-TUA6)-integrating [15,29] microtubules and GFP-tagged AtSYP51 [30,31] sorted as transmembrane protein to the tonoplast were expressed in transgenic Arabidopsis ecotype Columbia under the control of the CaMV 35S promoter.

Transgenic plantlets were grown from T2 seeds on sterile solid Murashige and Skoog basal medium (MS 1/2, 0.5% sucrose, 0.8% agar) under continuous light (about 120 uE m-2 sec-1) at 24 °C. Observed samples consisted of plantlets transferred to liquid medium, supplemented with extracts dissolved in DMSO at the concentrations equivalent to those able to generate diversified effects on human cancer cells in vitro. In the case of low doses, the concentration was selected as the minimal concentration generating diversified effects, while the concentration of the high dose was selected close to the maximum dose tested. Plants were then tested in multiwell plates 6 days after germination and monitored in the following 18 h.

Full plantlets, to reduce stress, were mounted for fluorescence microscopical observation in water under glass coverslips, and hypocotyls were imaged using a confocal laser microscope LSM 710 (Carl Zeiss MicroImaging GmbH, Germany). GFP markers were detected in the wavelength range of 505–530 nm and thus were assigned the green color, while chlorophyll autofluorescence was detected above 650 nm, thus being assigned the blue color. An excitation wavelength of 488 was used.

### 4.16. Statistical Analysis

A non-paired *t* test was used to compare two groups, and a one-way analysis of variance (ANOVA) was used to compare between groups and determine significant differences. A *p*-value < 0.05 was employed in all tests. GraphPad Prism 8 was employed to perform the statistical analyses (GraphPad Software Inc., San Diego, CA, USA).

## Figures and Tables

**Figure 1 ijms-24-14920-f001:**
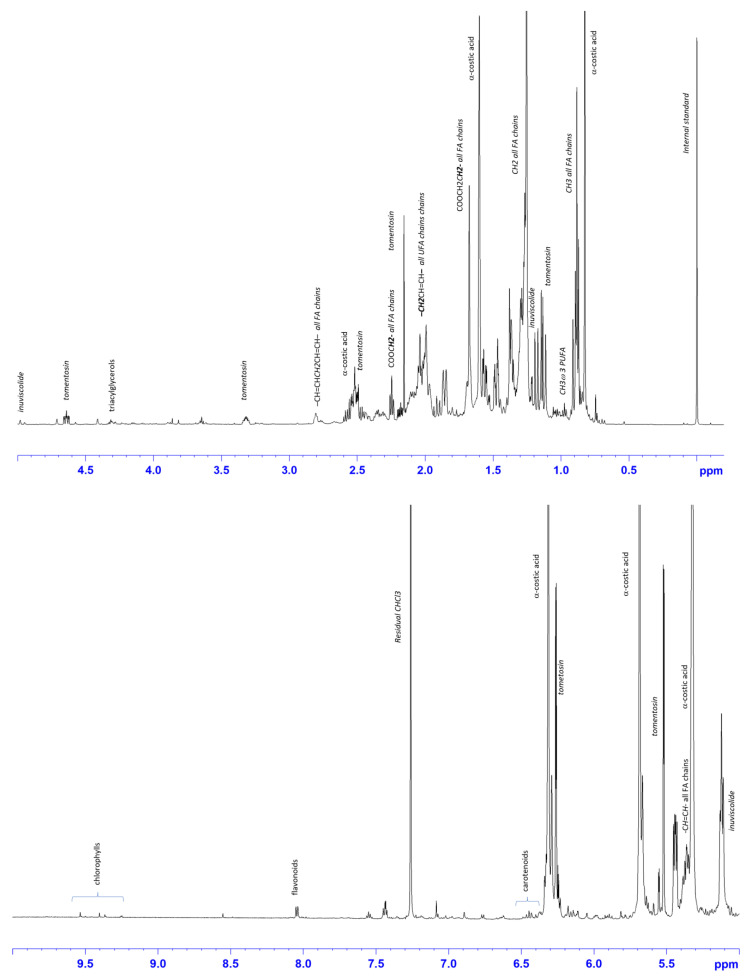
Expansions of the ^1^H-NMR *D. viscosa* hexanic spectrum (CDCl3). Diagnostic peaks of assigned metabolites are indicated. FA—fatty acids; PUFA—polyunsaturated fatty acids.

**Figure 2 ijms-24-14920-f002:**
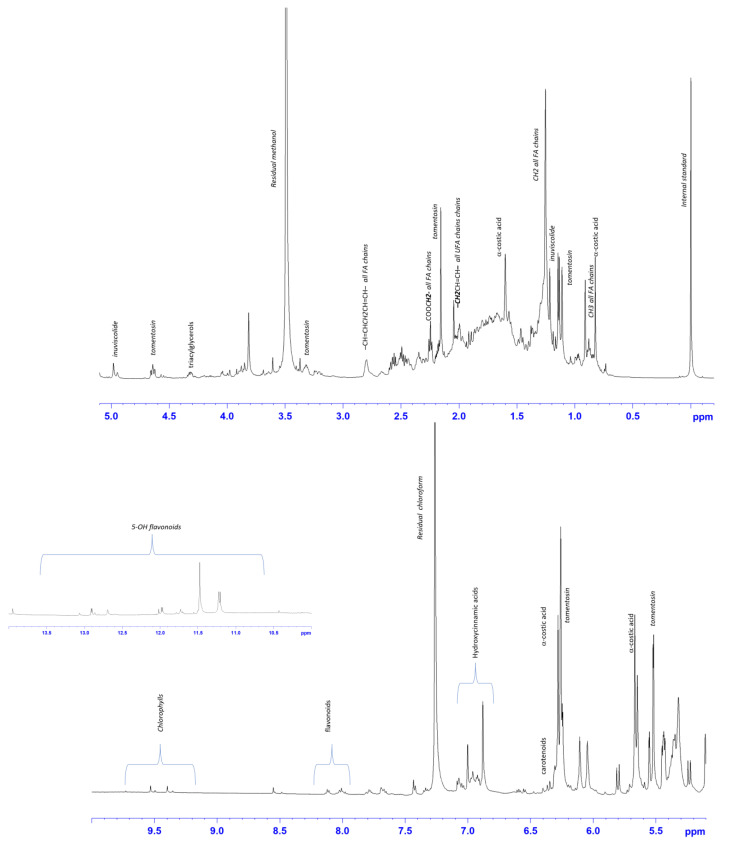
Expansions of the ^1^H-NMR *D. viscosa* methanolic spectrum (CDCl_3_). Portion of expanded downfield spectral region (10–14 ppm) is shown. Diagnostic peaks of assigned metabolites are indicated.

**Figure 3 ijms-24-14920-f003:**
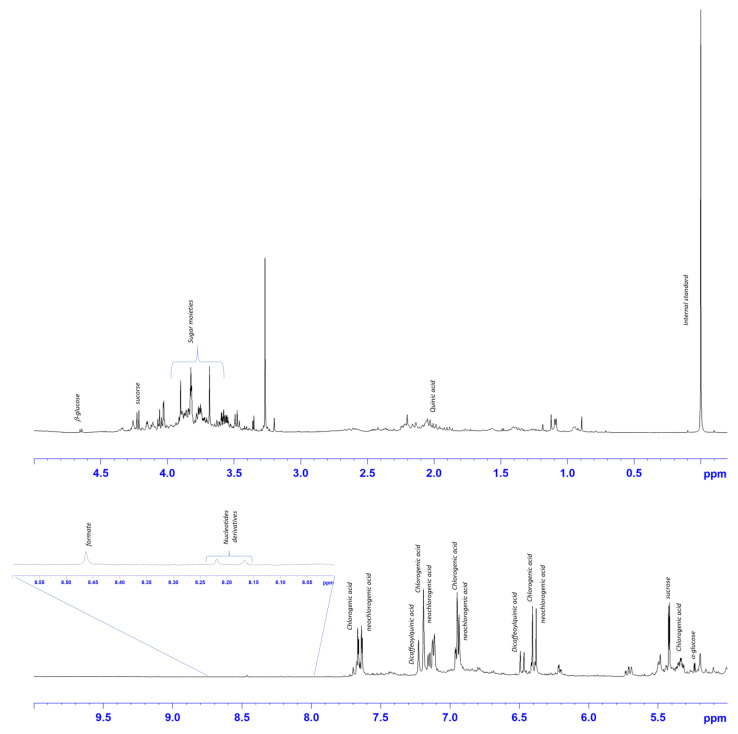
Expansions of the ^1^H-NMR *D. viscosa* aqueous spectrum (D_2_O). Portion of expanded downfield spectral region (8–8.60 ppm) is shown. Diagnostic peaks of assigned metabolites are indicated.

**Figure 4 ijms-24-14920-f004:**
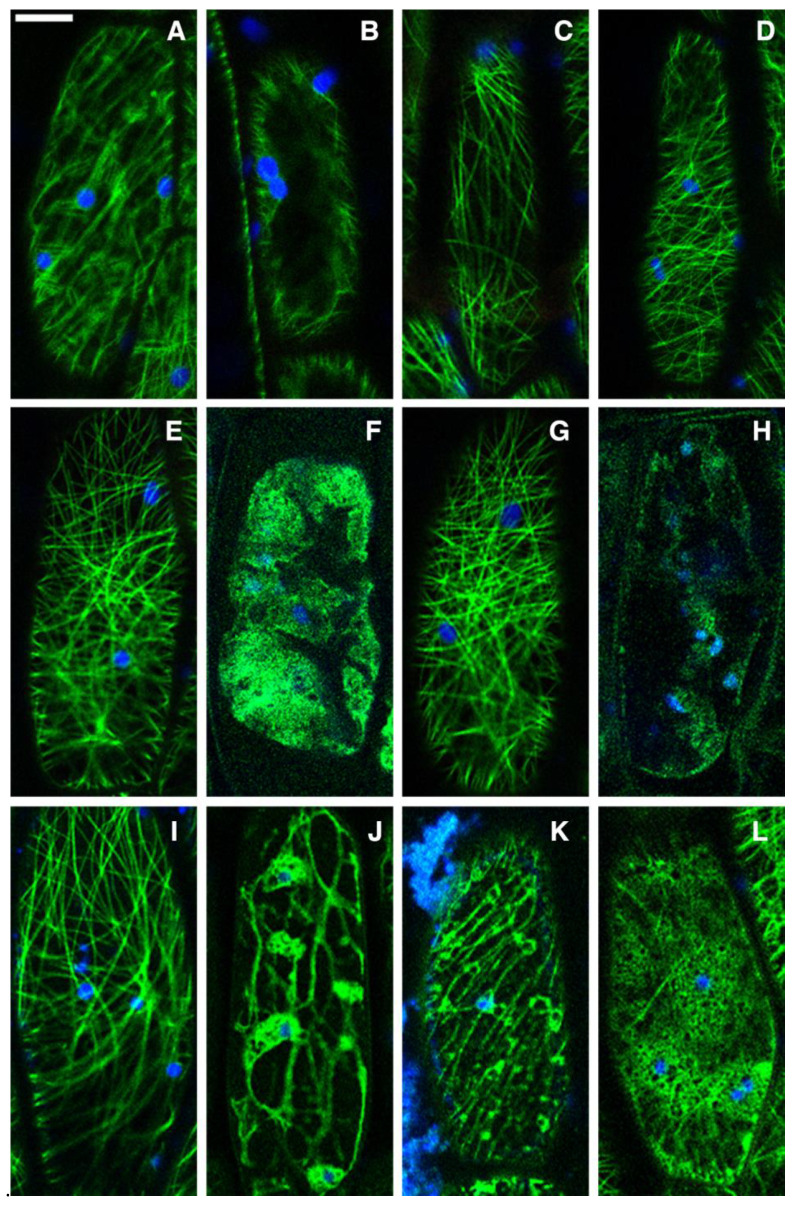
Confocal images of hypocotyl cells of *A. thaliana* stably expressing GFP-TUA6. Normal distribution of labelled microtubules in controls (**A**,**B**) with DMSO (**C**) and after high-dose AE treatment for 18 h (**D**). No effects of HE at low dose after 1 h (**E**) and collapse of cell after 18 h (**F**). Similarly, no effect observed with HE at high dose after 1 h (**G**), but the cell collapse after 18 h (**H**). No effects of ME at low dose after 1 h (**I**) and evident effects after 18 h (**J**). Evident effects of ME at high dose already in 1 h (**K**), increasing after 18 h (**L**). In blue the epifluorescence of chlorophyll. Scale bar: 10 µm.

**Figure 5 ijms-24-14920-f005:**
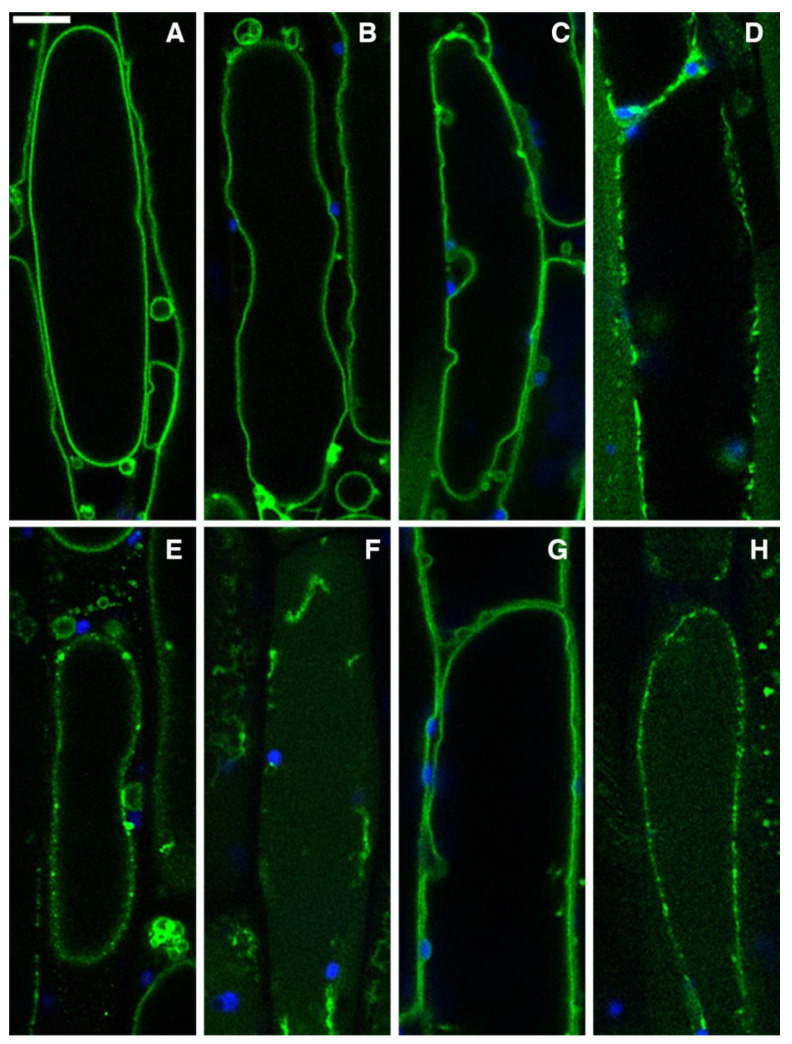
Confocal images of hypocotyl cells of *A. thaliana* stably expressing GFP::SYP51. The protein labels the tonoplast normal pattern in controls (**A**) with DMSO (**B**) and after high-dose AE treatment for 1 h (**C**). AE induced stress after 18 h (**D**). Immediate effect on the tonoplast of HE at low dosage within 1 h (**E**), increasing after 18 h (**F**). No effect with ME at high dose after 1 h (**G**) and alteration of tonoplast after 18 h (**H**). In blue the epifluorescence of chlorophyll. Scale bar: 10 µm.

**Figure 6 ijms-24-14920-f006:**
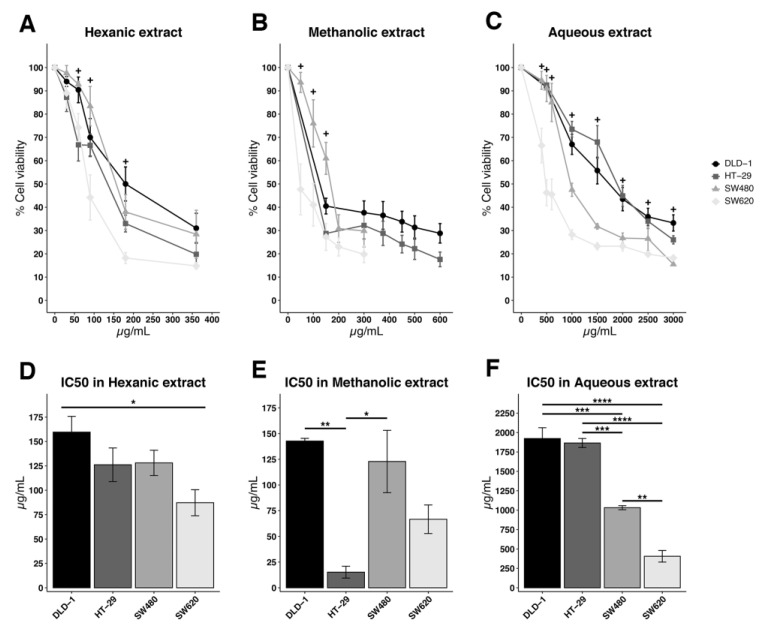
Colorectal cancer cells exposed to different extracts of *D. viscosa*. DLD-1, HT-29, SW480, and SW620 cell lines were exposed to hexanic (**A**), methanolic (**B**) and aqueous extracts (**C**) of *D. viscosa* and compared to different concentrations with cell viability assay. Then, another decrease at high concentrations of the extracts was shown. The presence of a significative different cell viability is indicated with “+”. Specific *p*-values of significant differences between the cells lines are reported in Appendix A. The IC50 value, valid respectively for (**A**–**C**) is reported in (**D**–**F**).The significant differences are indicated as **** *p* < 0.0001, *** *p* < 0.001, ** *p* < 0.01, and * *p* < 0.05. Numerical values are displayed in Appendix A.

**Table 1 ijms-24-14920-t001:** Main classes of molecules found in the three extracts.

Compounds	Hexanic (HE)	Methanolic (ME)	Aqueous (AE)
Soluble phenols (mg GAE/g dw)	11.37 ± 0.36	139.66 ± 7.21	168.16 ± 6.83
Flavonoids (mg CE/g dw)	39.01 ± 0.40	171.37 ± 8.99	207.54 ± 2.91
Condensed tannins (mg CE/g dw)	8.04 ± 0.17	4.08 ± 0.02	0.67 ± 0.05
Ascorbic acid (mg/g dw)	0.96 ± 0.02	4.81 ± 0.07	13.97 ± 0.03
Dehydroxyascorbic acid (mg/g dw)	0.39 ± 0.11	0.24 ± 0.02	1.14 ± 0.16
Total vitamin C (mg/g dw)	1.35 ± 0.13	5.05 ± 0.09	15.11 ± 0.19
Chlorophyll a (mg/g dw)	0.866 ± 0.008	1.384 ± 0.006	0.050 ± 0.000
Chlorophyll b (mg/g dw)	0	0.142 ± 0.008	0.047 ± 0.001
Total carotenoids (mg/g dw)	0.887 ± 0.006	0.239 ± 0.003	0.014 ± 0.000

**Table 2 ijms-24-14920-t002:** Antioxidant activity of the three extracts.

Antioxidant Activity	(µmol Trolox Equivalents/g dw)	ABTS+ Assay IC50 (µg/mL)
Hexanic (HE)	46.75 ± 0.25	256.71 ± 1.37
Methanolic (ME)	1503.22 ± 18.89	7.97 ± 0.10
Aqueous (AE)	2116.76 ± 30.75	5.66 ± 0.08

**Table 3 ijms-24-14920-t003:** HPLC analysis of polyphenols and isoprenoids (Tocochromanols, carotenoids, and chlorophylls); concentration per g of extract.

Bioactive Compounds		HE	ME	AE
**Polyphenols** (mg/g extracts)	Chlorogenic acid	0.78	4.06	22.90
	Di-*O*-caffeoylquinic acid	3.40	60.38	50.93
	Di*-O*-caffeoylquinic acid isomer	3.04	38.87	55.74
	Rosmarinic acid	0.02	4.05	7.77
	**Total**	**7.24**	**107.36**	**137.34**
**Tococromanols** (µg/g extract)	α T3	270.06	nd	nd
	β T	315.66	nd	nd
	α T	1178.72	nd	nd
	**Total**	**1764.44**	**nd**	**nd**
**Carotenoids** (µg/g extract)	Lutein	129.88	227.53	nd
	β carotene	993.19	nd	nd
	9 cis β carotene	155.91	nd	nd
	**Total**	**1278.98**	**227.53**	**nd**
**Chlorophyll b** (µg/g extract)		**115.45**	**708.57**	**nd**

**Table 4 ijms-24-14920-t004:** Percentage of fatty acids detected in the three extracts through GC-MS analysis.

Fatty Acids (%)	HE	ME	AE
Myristic acid (C14:0)	13.14	1.92	nd
Palmitic acid (C16:0)	22.67	18.59	nd
Stearic acid (C18:0)	0.57	0.66	nd
Arachidic acid (C20:0)	0.37	0.47	nd
Behenic acid (C22:0)	1.67	2.12	nd
Palmitoleic acid (C16:1)	1.31	3.32	nd
Oleic acid (C18:1 n-9c)	8.61	2.39	nd
11-Octadecenoic acid (C18:1 n-7c)	4.47	2.79	nd
cis-13-Eicosenoic acid (C20:1 n-7c)	2.56	0.69	nd
Linoleic acid (C18:2 n-6)	16.15	17.04	nd
Linolenic acid (C18:3 n-3)	28.49	49.99	nd
SFA	39.73	23.69	nd
MUFA	15.64	9.19	nd
PUFA	44.64	67.03	nd

**Table 5 ijms-24-14920-t005:** Morphological and molecular characteristics of the different cell types used in this work, including the consensus molecular subtype (CMS) in which they are usually classified. The principal proteins are shown in the wild-type (wt), over-expressed (up), down-expressed (down), and the specific mutations.

	DLD-1	HT29	SW480	SW620
**CMS**	CMS1	CMS3	CMS4	CMS4
**Morphology**	undifferentiated	colon-like cells	undifferentiated	undifferentiated
**Appearance**	mesenchymal	epithelial-like	mesenchymal	mesenchymal
**MSI**	MSI	MSS	MSS	MSS
**CIMP**	CIMP+	CIMP+	CIMP-	CIMP-
**CIN**	>>>	<	<	<
**TP53**	p.S241F	p.R273H	p.R273H; p.P309S	p.R273H; p.P309S
**KRAS**	p.G13D	wt	p.G12V	p.G12V
**BRAF**	wt	p.V600E; p.T119Sc	wt	wt
**PIK3CA**	p.E545K; p.D549N	wt	wt	wt
**SNAI1**	normal	normal	normal	up
**MSH5**	down	wt	wt	wt
**KI67**	up	up	down	up
**CDH1/E cadherin**	normal	normal	normal	down

## Data Availability

Data are available on request from the authors.

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
