# Peer review of "Methanolic Extracts of D. viscosa Specifically Affect the Cytoskeleton and Exert an Antiproliferative Effect on Human Colorectal Cancer Cell Lines, According to Their Proliferation Rate"

_ijms, 2023, doi:10.3390/ijms241914920_

Round 1
Reviewer 1 Report
In this study, the authors intended to validate a prescreening approach to identify the fraction of Dittrichia viscosa, which exhibits the most apparent antiproliferative and anticancer effects. D. viscosa, a perennial plant, exhibits pharmacological effects, including antioxidant, cytotoxic, antiproliferative, and anticancer effects. The authors performed an innovative prescreening using a transgenic plant screening method and tested different extract fractions on cell proliferation of cell lines of colorectal cancer.
The subject is interesting, the manuscript is well-structured, and presents new and valuable results that be of great interest to the researchers in the field. However, some parts can be improved in terms of writing and clarity.
Below are the comments which need to be addressed:
- In Line 303, the authors described that tubulin is considered a strategic molecular target for antitumor drugs, and tubulin inhibitors are supposed to be less prone to develop multi-drug resistance (MDR) compared to other metabolites. However, it is unclear whether the changes in the cytoskeleton induced by the extracts are because the cytoskeleton is a direct target (acting as direct inhibitors) of those fractions or because of the induction of the cell death process. Especially because extracts usually have pleiotropic effects. Please explain.
- Similarly, in the discussion section, the authors described that the pre-screening of the three extract fractions found that ME had a quite evident effect on microtubule polymerization without significantly affecting the cell compartmentalization. However, in line 192, they described cell collapse and lethality. Please explain.
- The authors must explain how the concentration used in the cytoskeleton screening methods was selected.
- In the conclusions section, I suggest adding future directions of the study as well as what were the limitations to your study.
Author Response
We thank reviewer one for the valuable comments. These highlight defects in our manuscripts that do not allow us to clearly express what we wanted to communicate. Text was revised extensively.
COMMENT: In Line 303, the authors described that tubulin is considered a strategic molecular target for antitumor drugs, and tubulin inhibitors are supposed to be less prone to develop multi-drug resistance (MDR) compared to other metabolites. However, it is unclear whether the changes in the cytoskeleton induced by the extracts are because the cytoskeleton is a direct target (acting as direct inhibitors) of those fractions or because of the induction of the cell death process. Especially because extracts usually have pleiotropic effects. Please explain.
REPLY: Text was perfected underlying that tubulin correct assembling is essential for the formation of the mitotic spindle. This is essential to qualify tubulin as strategic target. We believe to evidence in this work the direct effects on cytoskeleton. The indirect cytoskeleton remodeling because of PCD require time which exceed the shorter terms of the experiments proposed. Moreover, the endomembrane marker we used would evidence the events associated to cell death before the appearance of effects on tubulin. This was indeed the case for the HE, as shown in figure 5E. By our observation of both cytoskeleton and endomembranes, we can indeed discriminate between direct (visible within short time: Fig4K vs Fig 5G) and indirect effects (visible only after prolonged exposition: Fig4G vs Fig5E, F).
A pleiotropic effect of the extracts cannot be ruled out, and above all a non-specific cell death inducing effect affecting healthy cells as well as tumor cells, just like the cytostatics normally used as chemotherapeutics. As with current chemotherapy, this effect is more damaging to dividing cells, hence the applicability of neoplastic therapies against rapidly dividing abundant cells characteristic of the tumor mass, and the certain protection recorded for slower dividing cells. This also applies to our extracts but pre-screening on transgenic plants helps to increase the chance to focus research on compounds targeting cytoskeleton first.
We now mentioned these figures details in the discussion to help the readers.
COMMENT: Similarly, in the discussion section, the authors described that the pre-screening of the three extract fractions found that ME had a quite evident effect on microtubule polymerization without significantly affecting the cell compartmentalization. However, in line 192, they described cell collapse and lethality. Please explain.
REPLY: ME have visible effects on cytoskeleton before any effect on endomembranes is detected. We interpreted this as an evidence that effects on endomembranes derive from cytoskeleton defects. We modified text from line 310-324: Our pre-screening analyzed the three extract fractions at low and high concentrations. The low dose was chosen because able to induce diversified effects on tumor cell lines, while the high dose was chosen at the limit of practical use, able to eventually overcome problems related to plant cell wall barrier. We found that ME had a quite evident effect on microtubules polymerization without affecting significantly the cell compartmentalization in the short period. The effect on endomembranes derives from cytoskeleton disruption and require time, leading to cell death in plants as, we hypothesize, in tumor cells. In other words, when cytoskeleton remodeling is observed first, the endomembranes alteration is a consequence leading to PCD. It means that the higher effect would be on proliferating cells affecting the mitotic bundle. If endomembranes are altered before or at the same time with cytoskeleton, the molecular target of the extract is unpredictable and may easily cause off target effect in human cells.
ME high dose have a clear effect on cytoskeleton within 1 h (Figure 4K), without affecting endomembranes (Figure 5G), on the contrary HE has no effect on cytoskeleton even at high dose (Figure 4G) but strongly affects endomembranes (Figure 5E, F).
COMMENT: The authors must explain how the concentration used in the cytoskeleton screening methods was selected.
REPLY: A better explanation was inserted in the method description, line 587-590: Observed samples consisted of plantlets transferred to liquid medium, supplemented with extracts dissolved in DMSO at the concentrations equivalent to those able to generate diversified effects on human cancer cells in vitro. In the case of low dose, the concentration was selected as the minimal concentration generating diversified effect and in the case of high dose it was selected a dose close the maximum tested. Plants were then testes into multiwell plates 6 days after germination and monitored in the following 18h.
COMMENT: In the conclusions section, I suggest adding future directions of the study as well as what were the limitations to your study.
REPLY: We modified the final part of discussion. This is our third validation of the use of transgenic plants as support in the characterization of subcellular effects of complexes or single molecules (VERGARA et al. 2015; PAPADIA et al 2017) and we believe the method (Di Sansebastiano and Barozzi 2018) is ready for new challenges in real pre-screening.
Reviewer 2 Report
In the manuscript by Angana and Rojas et al., the authors examined the antioxidant, antiproliferative and anti-cancer effects of all three fractions of Dittrichia viscosa. Among them the methanolic extract exhibits the most promising antiproliferative and anticancer effects with reduced off-target effects.
Although the data presented in the manuscript is interesting, it needs extensive revision.
I have a few major/minor suggestions for polishing the manuscript.
Major Comments:
1. The manuscript needs to be checked by a native English speaker.
2. Page 3, the antioxidant effect was shown using a tabular form. The authors should depict it graphically as it is a very important data of the manuscript. Calculate the IC50. Are the data given in the table IC50 values?
3. Page 8, Line 183 ‘what does ‘extracts on cell biology’ mean? Give proper explanation. Also, the extracts were tested on transgenic hypocotyl cells. The data the authors showed in plants necessarily does not mean that the same thing can be replicated on cancer cells’. The authors should explain this experiment.
4. In Figure 3, the authors stated that tubulin can be targeted by the extracts of D.viscosa. Tubulin is a housekeeping gene. If somebody used tubulin as a target for anti-cancer agent, it would create tremendous side-effects. The authors should explain the rationale behind this experiment. Page 10, Line 212, the authors said there is probably a strong induction of autophagy and stress related multivesicular bodies formation’ Are there any evidence for this observation or the authors have done any experiments? Explain this statement.
5. Page 10, Line 208, ‘GFP-tagged AtSYP51, GFP:SYP51, is sorted as the related QcSNARE and sorted as transmembrane protein to the tonoplast’. Explain this sentence.
6. The results of Figure 5 seem confusing. The authors should rewrite it in a simple way.
7. The authors should study the effect of the extracts on normal cell lines. The comparison conducted across different cancer cell lines does not provide sufficient evidence to determine the side effects associated with the extracts.
8. The IC50 values from Figure 6 should be depicted in a tabular manner in Figure 6.
9. The authors have discussed potential mechanisms that can cause these anti-cancer effects. The authors tried to relate the mechanism with the molecular characteristics of the cell line. Without proper molecular biology studies, the authors cannot claim these probabilities. The authors should depict the mechanism through which the extracts can work on these cell lines. The authors should conduct additional experiments on this.
Minor Comments
1. Page 1, Line 44 ‘Plants are widely……cancer’. Give proper reference.
2. Page 1, Line 45 ‘Medicinal plants are highly….off target effects’. Give proper reference.
3. Page 2 Line 54 ‘chemotherapics’ should be ‘chemotherapeutics’.
4. Page 2, Line 56 ‘effecs’ should be ‘effects’.
5. Page 2, Line 56, ‘specificic’ should be ‘specific’.
6. In the Introduction Section add the references of the previous literatures showing the effects of D.viscosa on cancer cells. There are references like
Mrid, R.B.; Bouchmaa, N.; Kabach, I.; Zouaoui, Z.; Chtibi, H.; Maadoudi, M.E.; Kounnoun, A.; Cacciola, F.; Majdoub, Y.O.E.; Mondello, L.; et al. Dittrichia viscosa L. Leaves: A Valuable Source of Bioactive Compounds with Multiple Pharmacological Effects. Molecules 2022, 27, 2108. https://doi.org/10.3390/molecules27072108
Vuko, E.; Dunkić, V.; Maravić, A.; Ruščić, M.; Nazlić, M.; Radan, M.; Ljubenkov, I.; Soldo, B.; Fredotović, Ž. Not Only a Weed Plant—Biological Activities of Essential Oil and Hydrosol of Dittrichia viscosa (L.) Greuter. Plants 2021, 10, 1837. https://doi.org/10.3390/plants10091837.
Cite these references.
7. Colorectal Cancers should be elaborately described in the Introduction Section. Also add why this research is useful over existing therapies of colorectal cancer.
8. Page 2, Line 74, ‘the specific effect of a specific D.viscosa’- Remove the first specific.
9. There are lots of grammatical errors in the Introduction section. Correct those errors.
10. Check the spellings of tococromanols and clorogenic acid In Page 7.
11. Page 10, Line 208, ‘GFP-tagged AtSYP51, GFP::SYP51, is sorted as the related QcSNARE and sorted as transmembrane protein to the tonoplast’. Explain this sentence.
12. Line 227-229 is a repetitive sentence. The lines were also used in the abstract section. The authors should rephrase the sentences.
13. Table 5 is not related to this study. It just shows the morphological and molecular characteristics. They can include it in the supplementary figure.
14. Page 12, Line 242 ‘compare’ will be ‘comparative’.
1. The manuscript needs to be checked by a native English speaker.
2. There are lots of grammatical errors in the Introduction section. Correct those errors.
Author Response
We thank reviewer 2 for the valuable comments. In some cases we can not fully satisfy the concerns because we have the impression that the attention of the study to the colon cancer and its therapies was overestimated by the reviewer. Some requests were beyond the scope of our work but we understood that we were not clear in the explanations, so we perfected the text in several parts, dealing with all the indicated concerns.
Major Comments:
COMMENT 1: The manuscript needs to be checked by a native English speaker.
REPLY: It was done
COMMENT 2: Page 3, the antioxidant effect was shown using a tabular form. The authors should depict it graphically as it is a very important data of the manuscript. Calculate the IC50. Are the data given in the table IC50 values?
REPLY: Indeed, we agree with the reviewer on the importance of the data of the antioxidant effect of the extract, but we consider that this importance is highlighted and it is appropriate to keep the format of expression of these data coinciding with the format usually used in the literature to refer to the potential antioxidant power of an extract, which is usually massively in tabular form referring to the universal unit of concentration (Trolox UM). It was not an IC50 since we were not talking about a cellular control effect, but rather this information refers to the antioxidant capacity that the extracts also possess, which can be beneficial as an adjuvant in future treatments, and it is the way of expressing it that allows comparison with other international publications in the field of results referring to antioxidant power. Then we decided to enrich the table with both, somehow complementary, values.
|
Antioxidant activity |
(µmol Trolox equivalents/g dw) |
ABTS+ assay IC50 (µg/ml) |
|
Hexanic (HE) |
46.75 ± 0.25 |
256.71 ± 1.37 |
|
Methanolic (ME) Aqueous (AE) |
1503.22 ± 18.89 2116.76 ± 30.75 |
7.97 ± 0.10 5.66 ± 0.08 |
COMMENT 3: Page 8, Line 183 ‘what does ‘extracts on cell biology’ mean? Give proper Also, the extracts were tested on transgenic hypocotyl cells. The data the authors showed in plants necessarily does not mean that the same thing can be replicated on cancer cells’. The authors should explain this experiment.
REPLY: We reformulated the text (lines 183-188) to clarify we investigate the effects on the biology of the cell observing the markers. We addet the following text: “These markers label important subcellular structures, differently organized in plant cell but based on molecular processes largely conserved with animal and human cells (Vergara et al. 2015).”. In the introduction and in the discussion we suggest that this effect can be predictive of what will be evidenced in cancer cell with much higher difficulty and cost.
COMMENT 4: In Figure 3, the authors stated that tubulin can be targeted by the extracts of D.viscosa. Tubulin is a housekeeping gene. If somebody used tubulin as a target for anti-cancer agent, it would create tremendous side-effects. The authors should explain the rationale behind this experiment.
REPLY: This is a misunderstanding due to the common consideration of genes as target of chemotherapeutics. We refer to the protein as target and this is a reason why we expect less off target effect. As with taxol/paclitaxel, only where tubulin is remodelling actively (for example on forming mitotic bundle) the effect is stronger.
COMMENT 5: Page 10, Line 212, the authors said there is probably a strong induction of autophagy and stress related multivesicular bodies formation’ Are there any evidence for this observation or the authors have done any experiments? Explain this statement.
REPLY: We believe that the strenght of the proposed approach is in simplycity. Our statement on authophagy and multivesicular bodies formation is supported by what we observe “since these are the processes able to internalize and solubilize in the vacuoles a membrane-anchored protein such as GFP::SYP51 (De Benedictis et al. 2013).” Text was edited from line 216 to 218.
COMMENT 6: Page 10, Line 208, ‘GFP-tagged AtSYP51, GFP:SYP51, is sorted as the related QcSNARE and sorted as transmembrane protein to the tonoplast’. Explain this sentence.
REPLY: At line 211 we now precise that it , is a TGN and tonoplast marker. Then we explain how it is sorted.
COMMENT 7: The results of Figure 5 seem confusing. The authors should rewrite it in a simple way.
REPLY: Lines 211-230 we improved figure 5 description
COMMENT 8: The authors should study the effect of the extracts on normal cell lines. The comparison conducted across different cancer cell lines does not provide sufficient evidence to determine the side effects associated with the extracts.
REPLY: The side effects will be studied in comparison with those produced by current chemotherapeutics when we are in the experimental animal study step. Cell viability of normal cells in culture compromised by chemotherapy does not invalidate its use in patients since it is the dose of use that marks the side effects in complex organisms. On the other hand, the use at the cellular level of different tumor lines does provide information regarding the applicability of the D. viscosa extract and its future extracts in combined treatments in different types of tumors and phases of metastatic disease, where a primary tumor can metastasize in organs very different from the primary detection.
Morover, the main goal of the study was to porpose a pre-screening approach on transgenic plants and the tumor lines are sued to support the preliminary observation. The characterization of off-target effect is one of those complex and difficult task that pre-screening aim to reduce to fewer cases. It is anyhow beyond the scope of this study.
COMMENT 9: The IC50 values from Figure 6 should be depicted in a tabular manner in Figure 6.
REPLY: We invite the reviewer to reconsider the request since we selected this representation to allow an easyer reading of the data and the relative significance for the different cell lines. A table would not allow such a rapid visualization of significant differences.
COMMENT 10: The authors have discussed potential mechanisms that can cause these anti-cancer effects. The authors tried to relate the mechanism with the molecular characteristics of the cell line. Without proper molecular biology studies, the authors cannot claim these probabilities. The authors should depict the mechanism through which the extracts can work on these cell lines. The authors should conduct additional experiments on this.
REPLY: We see the point of the reviewer and it would certainly be interesting but this is beyond the scope of our work. Here we discussed about the possibility to relate the effect of the extracts to proliferation having in mind the evidence of the effect on microtubules observed in the pre-screening in transgenic plants.
Thanks to pre-screening efforts, further investigation in tumor cell lines, could be concentrated on the effect of methanolic extract with better chances of success.
Minor Comments
- Page 1, Line 44 ‘Plants are widely……cancer’. Give proper reference. done
- Page 1, Line 45 ‘Medicinal plants are highly….off target effects’. Give proper reference. 5 references were added
- Page 2 Line 54 ‘chemotherapics’ should be ‘chemotherapeutics’. done
- Page 2, Line 56 ‘effecs’ should be ‘effects’. done
- Page 2, Line 56, ‘specificic’ should be ‘specific’. done
- In the Introduction Section add the references of the previous literatures showing the effects of D.viscosaon cancer cells. There are references like
Mrid, R.B.; Bouchmaa, N.; Kabach, I.; Zouaoui, Z.; Chtibi, H.; Maadoudi, M.E.; Kounnoun, A.; Cacciola, F.; Majdoub, Y.O.E.; Mondello, L.; et al. Dittrichia viscosa L. Leaves: A Valuable Source of Bioactive Compounds with Multiple Pharmacological Effects. Molecules 2022, 27, 2108. https://doi.org/10.3390/molecules27072108
Vuko, E.; Dunkić, V.; Maravić, A.; Ruščić, M.; Nazlić, M.; Radan, M.; Ljubenkov, I.; Soldo, B.; Fredotović, Ž. Not Only a Weed Plant—Biological Activities of Essential Oil and Hydrosol of Dittrichia viscosa (L.) Greuter. Plants 2021, 10, 1837. https://doi.org/10.3390/plants10091837.
Cite these references.
Thank you for the indication. Indeed that part was not sufficiently supported.
- Colorectal Cancers should be elaborately described in the Introduction Section. Also add why this research is useful over existing therapies of colorectal cancer.
Line 77 to 86 were inserted to clarify that CRC cell lines were used as models of primary carcinomas for preclinical studies and, in this case, to support the observation done on transgenic plants. We have no ambition to suggest therapies but a new approach to natural product screening.
- Page 2, Line 74, ‘the specific effect of a specific D.viscosa’- Remove the first specific. done
- There are lots of grammatical errors in the Introduction section. Correct those errors.
- Check the spellings of tococromanols and clorogenic acid In Page 7. Thank you for the correction of Chlorogenic, written in different ways in the text.
- Page 10, Line 208, ‘GFP-tagged AtSYP51, GFP::SYP51, is sorted as the related QcSNARE and sorted as transmembrane protein to the tonoplast’. Explain this sentence.
Lines 220-223 should now be more self explanatory.
- Line 227-229 is a repetitive sentence. The lines were also used in the abstract section. The authors should rephrase the sentences.
Indeed, it was redundant. We rephrased in the abstract: “. In our study we used HT-29 colon-like cell line and three cell lines showing positive regulation of epithelial-mesenchymal transition and TGFβ signatures ( DLD-1, SW480 and SW620), representative of different stages of colorectal cancer.”
- Table 5 is not related to this study. It just shows the morphological and molecular characteristics. They can include it in the supplementary figure.
It is true that this is not a relevant result but a valuable description of the used cell line. We hope to satisfy the reviewer moving the table in the method sectio: 5.12. Cell lines and cell culture
- Page 12, Line 242 ‘compare’ will be ‘comparative’. done
English was revised by a native English speaker.
Reviewer 3 Report
Reviewer comments and suggestions
The authors in this study identified the fraction exhibiting the most promising antiproliferative and anticancer effects of Dittrichia viscosa. In particular, we investigated the effect of different extract fractions of Dittrichia viscosa on the cytoskeleton using a transgenic plants screening method. Tumors are heterogeneous and cytoskeleton, tubulin in particular, is considered a strategic target for antitumor drugs. The authors used colon-like cell lines characterized by expression of gastrointestinal differentiation markers (as HT-29 cell line) and undifferentiated cell lines showing positive regulation of epithelial-mesenchymal transition and TGFβ signatures (as DLD-1, SW480 and SW620 cell lines). They showed that all of the 3 D. viscosa extract fractions, have antiproliferative effect but the pre-screening on transgenic plants anticipated that the methanolic fraction may be the most promising, targeting the cytoskeleton specifically and possibly resulting in fewer side effects.
Overall, the manuscript was well written. However, a few concerns/comments needed to be explained/modified.
- Line 2 D. viscosa should be italic
- Line 46-48 the authors could add references for these lines
- Two typographical errors are present in line 56
- Line 77-78 I could not understand the line, please explain it
- Comments for table 2 Why it was so low, can you explain
- Line 182 please explain hypocotyl cells
- Comments for Figure 4 (line 202–204) please indicate these in the figure if possible
- Line 235 it would be nice if they could mention the name of specific cell line used in the legend part along with explaining the abbreviations
- Line 270 many studies need more citations
- Line 273-274 if the results were yours so how the authors need to add a reference for this
- Line 279-280 please explain the lines and it if needed to add a table or figures at the respective place
- Line 321-313 Why the authors want to discuss patients
- All references should be modified based on MDPI.
Author Response
We thank reviewer one for the valuable comments. We dealt with all concerns trying to give explanations. Text was revised extensively.
- Line 2 D. viscosa should be italic
done
- Line 46-48 the authors could add references for these lines
5 references were added
- Two typographical errors are present in line 56
done
- Line 77-78 I could not understand the line, please explain it
Text was expanded to be more clear.
- Comments for table 2 Why it was so low, can you explain
These are normal values in sequential extraction. Similar data are reported in cited literature.
- Line 182 please explain hypocotyl cells
Hypocotyl is an anatomical part of the plant. We rephrased: “transgenic cells in the plant hypocotyl”
- Comments for Figure 4 (line 202–204) please indicate these in the figure if possible
We tried not to overload the legend of the figure that is already rich of panels. If acceptable we would prefer not to add comments in figures legend.
- Line 235 it would be nice if they could mention the name of specific cell line used in the legend part along with explaining the abbreviations
Sorry but we do not understand this request. In figure 6 the 4 cell lines are always treated in parallel. Their code correspond to the description now found in the method section.
- Line 270 many studies need more citations. Indeed, 3 references were inserted.
- Line 273-274 if the results were yours so how the authors need to add a reference for this. Sentence was rephrased
- Line 279-280 please explain the lines and it if needed to add a table or figures at the respective place
Text revision for the concerns of the other reviewers may have clarified the situation
- Line 321-313 Why the authors want to discuss patients
We do not discuss patients but simply underline that tumors develop in very heterogeneous environments that depends on patients. In line now 326 (because of added text) we modified the text to avoid too many references to patients.
- All references should be modified based on MDPI.
Round 2
Reviewer 2 Report
In the manuscript by Anglana and Rojas et al., the authors examined the antioxidant, antiproliferative and anti-cancer effects of all three fractions of Dittrichia viscosa. Among them the methanolic extract exhibits the most promising antiproliferative and anticancer effects with reduced off-target effects.
The authors have addressed all the comments. The manuscript can be accepted in its present form.